# Dynamic activation catalysts for CO$_2$ hydrogenation

Zhewei Zhang[1,2], Jun Yao[1,2], Chenyang Shen[1,2], Fengfeng Li[1], Changshun Deng[1], Taotao Zhao[1], Xuefeng Guo [1], Yan Zhu [1], Xiangke Guo[1], Nianhua Xue[1], Luming Peng [1] & Weiping Ding [1] ✉

In typical heterogeneous catalytic reactions, catalysts, whether fixed or flowing, maintained their bulk and surface structures as stable as possible. We report here dynamic activation catalysts having continuously generate highly active sites in working, which enables a usually low active Cu/Al$_2$O$_3$ catalyst for CO$_2$ hydrogenation, showing extraordinary catalytic performances. Using reaction streams in unusually high linear speed to blow and carry the Cu/Al$_2$O$_3$ particulates to collide cyclically with a rigid target, the CO$_2$ conversion rate is more than three times enhanced at methanol selectivity promoted to 95% from less than 40% and the methanol space-time-yield is six times increased. By experimental and theoretical investigation, the dynamic activation of Cu/Al$_2$O$_3$ is defined as a discrete condensed state with a distorted and elongated lattice, reduced coordination, and abnormal catalytic properties. We envision that continuous research on the dynamical activation catalysts will advance novel methods for promoting catalytic performance and discovering new catalytic reactions.

According to traditional theories, reaction rates of heterogeneous catalysis, primarily governed by factors such as temperature, the number of active sites of the catalyst, and reactant concentrations[1–3], were generally independent of reaction gas flow rate. The catalyst surface on which the catalytic sites were tried every means to maintain stability in the catalytic reactions. However, considering that catalytic processes are inherently dynamic, we propose to view catalysts from a perspective of motion, where no molecule collides with the same catalytic particle twice. Then, could we create catalysts that are constantly changing or activating during catalytic reactions? Mechanochemistry, such as ball milling, leverages high-energy collisions to drive complex reactions[4,5]. It generates localized hotspots (>1100 K) or defects that promote lattice restructuring or amorphization[6]. However, the energy input evolved in ball milling is too high and difficult to control, often triggering unwanted side reactions. We report here an innovative design of a reactor that harnesses low-power energy, such as ~0.34 W acting on 0.5–1 g catalyst, from the kinetic energy of reaction gas itself, without any mechanical moving parts, to effectively activate the catalyst surface, significantly boosting catalytic performance.

In this work, we introduce a novel mode of catalysis wherein the reaction rate significantly increases with the augmentation of reaction gas speed, challenging the traditional notion about the relationship of reaction rate with space velocity[7,8], due to the dynamic activation effect which creates more sites and/or sites more active, corresponding to elevation in the pre-exponential factor (A) and/or a reduction in the activation energy (Ea), thereby substantially enhancing the overall reaction rate or altering reaction mechanism. The concept is schematically described in Fig. 1, taking Cu/Al$_2$O$_3$ catalyst and the reaction of CO$_2$ hydrogenation to methanol as examples.

From a microscopic perspective, the heavy collision disrupts the quasi-steady-state approximation (QSSA) in conventional catalytic kinetics[9–11], which assumes a stable catalyst surface, low intermediate concentrations, stationary variations in concentrations ($d_{intermediate}/dt \approx 0$), and equalized rates of each step normalized by the stoichiometric numbers ($\sigma_i$)[12]. The current DAR system, through continuously

[1]Key Lab of Mesoscopic Chemistry, School of Chemistry and Chemical Engineering, Nanjing University, Nanjing 210023, China. [2]These authors contributed equally: Zhewei Zhang, Jun Yao, Chenyang Shen. ✉e-mail: Dingwp@nju.edu.cn

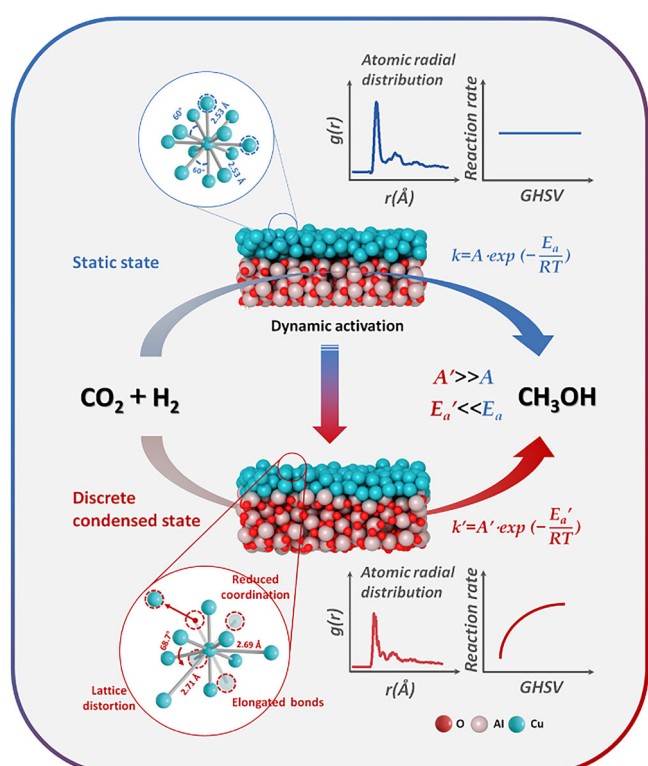

**Fig. 1 | Concept illustration.** Schematic view of the investigation on dynamic activation catalysts for $CO_2$ hydrogenation.

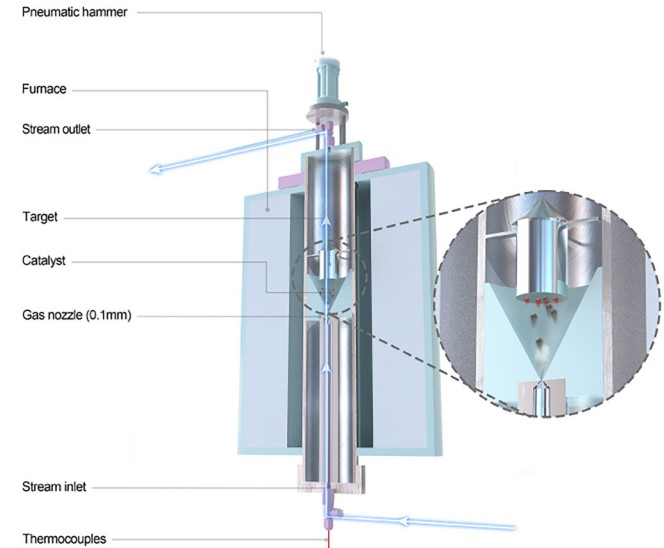

**Fig. 2 | Reactor design.** The reactor supplies continuous energetic collisions of catalyst particulates to dynamically activate the $Cu/Al_2O_3$ catalyst for $CO_2$ hydrogenation to methanol.

### Table 1 | The stripping energy of a Cu atom from $Cu/Al_2O_3$ in different thicknesses of copper on alumina

| Samples | Stripping energy* / eV |
|---|---|
| 20 wt.% $Cu/Al_2O_3$ | 1.12 |
| 40 wt.% $Cu/Al_2O_3$ | 0.97 |

*The stripping energy $E_{se}$ was calculated using the formula: $E_{se} = (E_N - E_{N-1})/n$, where $E_{se}$ represents the average energy required to remove a single atom from the system, N denotes the number of atomic layers, n is the number of atoms per layer, $E_N$ is the total energy of the system with N layers, and $E_{N-1}$ is the total energy of the system with N-1 layers.

evolving active sites under dynamic activation (accompanied by lattice deformation, reduced coordination number, and changes in crystallinity), primarily challenges the assumption of stationarity due to transient distortions leading to unstable intermediate dynamics and rate equalization, disrupted by mechanism shifts favoring the methanol pathway, thus necessitating reevaluation of these assumptions. These dynamic alterations not only change catalytic sites to inhibit sintering and agglomeration[13–15] but also modify the reaction mechanism, including to suppress side reactions that easily occur under traditional reaction conditions[16,17] and intensifying mass transfer and then catalytic efficiency[18–20]. With the concept developed in this work, generally low active $Cu/Al_2O_3$ for $CO_2$ hydrogenation shows extraordinary catalytic performance to methanol, comparable to or better than the best catalysts reported in the literature up to date.

## Results

### Scheme of dynamic activation

The dynamic activation catalysts were achieved using a purpose-made dynamic activation reactor, of which the structure is schematically shown in Fig. 2. It was a cylinder of 80 mm in diameter, made of 316 L stainless steel, connected to a cone at the bottom. A 0.1 mm diameter nozzle was installed at the bottom of the cone, and a rigid target, made of stainless steel, was installed 20 mm away from the nozzle. The catalyst powders were placed between the nozzle and the target. The gas mixture of $CO_2/3H_2$ was fed to the reactor through the nozzle. Under the operation conditions (P: 2.0 MPa; T: 300 °C; $3H_2/CO_2$, 360 ml min$^{-1}$), the distributions of gas flow field and the velocity of catalytic particulates were calculated through Fluent software using a discrete phase particle model simulation (Supplementary Fig. 1), yielding a nozzle exit gas velocity of ~452 m/s and a particle impact velocity on the target of ~75 m/s. The dynamic activation reactor employs a 0.1 mm nozzle to inject $CO_2/3H_2$ gas, blowing catalyst powder to continuously collide with a rigid target. An air hammer tapping the reactor every 3 s to prevent catalyst sticking and ensure cyclic impacts, as shown in Fig. 2.

The reaction tail gas was analyzed online by a gas chromatograph equipped with fa lame ionized detector (FID) and a thermal conductivity detector (TCD). The volume of the reactor was about 2 L and the gases in the reactor could be completely replaced within 120 min at a flow rate of 360 ml min$^{-1}$, as calibrated by $N_2$ (Supplementary Fig. 2). To some extent, the dynamic activation reaction (DAR) reported here reminds people thinking about Lanny Schmidt's research on fast flow reactor[21], which employs millisecond contact time reactors to enhance selectivity via high-velocity gas flow. However, the DAR focuses on the catalyst structural changes induced by collisions with suitable energy driven by the reaction gas itself, achieving superior catalytic performance.

### Catalytic performance

The catalysts 20 wt.% $Cu/Al_2O_3$ and 40 wt.% $Cu/Al_2O_3$ were prepared through impregnation using Cu $(NO_3)_2·6H_2O$ and nano γ-$Al_2O_3$ as raw materials. After drying, the catalysts were calcined at 450 °C in air to form $CuO/Al_2O_3$, which was then reduced to $Cu/Al_2O_3$ prior to reaction measurement. Considering the surface area of the nano alumina used (~150 m²·g$^{-1}$), the thickness of copper on the nano alumina is about 1-2 or 2–3 atomic layers for 20% or 40% $Cu/Al_2O_3$ (denoted as 20Cu or 40Cu in text), respectively. The basic information about the catalysts tested in this work are summarized in Supplementary Information (Supplementary Figs. 3, 4, Table 1, Supplementary Table 1).

Prior to the test, the original gases in the reactor were pumped out. A blank experiment without catalyst added was performed and the conversion of reaction gases was negligible (Supplementary Fig. 5). For the catalyst 40Cu, with multiple atomic layers of Cu on alumina, it exhibits a significant impact effect, as shown in Fig. 3a, due to the lower

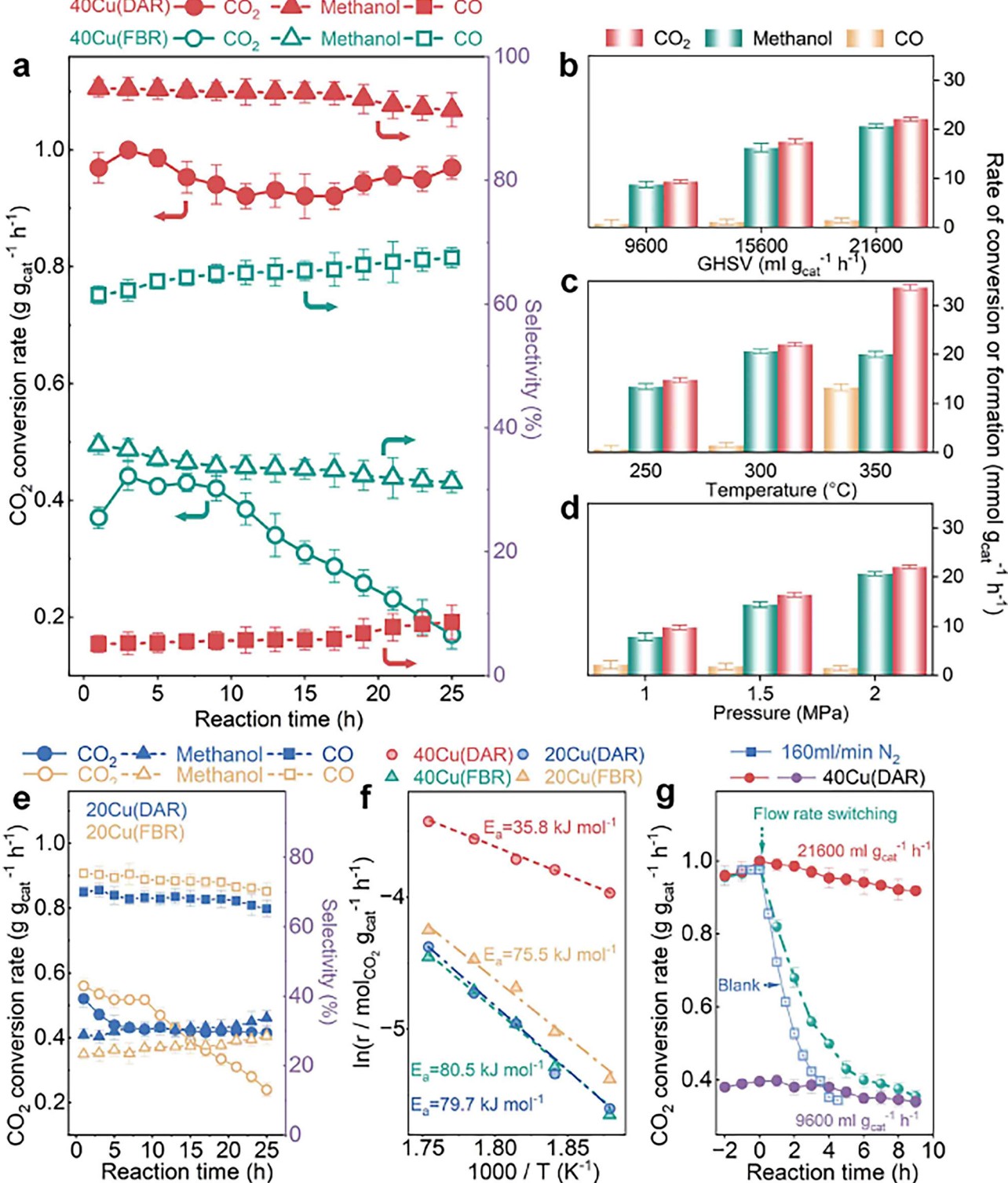

**Fig. 3 | Catalytic Performance. a** Catalytic performances of 40Cu under dynamic activation and in a traditional fixed bed. **b**, **c** and **d** respectively showing the effects of gas hour space velocity, temperature, and pressure on rates of $CO_2$ conversion, formation of methanol or CO ($g \cdot g_{cat}^{-1} \cdot h^{-1}$) over 40 Cu under dynamic activation. **e** Catalytic performance of 20Cu under dynamic activation and

traditional fixed-bed reaction. **f** Apparent activation energy (Ea) of $CO_2$ conversion over 40Cu. **g** Transition of $CO_2$ hydrogenation over 40Cu switched from GHSV = 21600 to 9600 $ml \cdot g_{cat}^{-1} \cdot h^{-1}$. (Cat.: 1 g; P: 2.0 MPa; T: 300 °C; $3H_2/CO_2$, GHSV = 21600 $ml \cdot g_{cat}^{-1} \cdot h^{-1}$, if unspecified). The error bars represent the standard deviation from three independent measurements.

stripping energy required among copper atoms (Table 1), considering the Tamm temperature of copper is very low[22]. The space-time-yield of methanol over 40Cu under dynamic activation reaction (DAR) reaches as high as 660 $mg \cdot g_{cat}^{-1} \cdot h^{-1}$, six times higher than that of static reaction in a traditional fixed bed reactor (~100 $mg \cdot g_{cat}^{-1} \cdot h^{-1}$, Supplementary

Fig. 6). The latter is named as fixed bed reaction (FBR) in context. Noteworthy, the selectivity to CO drastically decreases to ~5% from ~60%, indicating a significant change in the surface properties of the catalyst upon dynamic activation. In addition, the catalyst shows relatively better stability on stream in DAR. When the gas flow rate is

adjusted to 160 ml·min$^{-1}$, interestingly, the impact effect has decreased significantly (Supplementary Fig. 7), implying the impact energy must be larger than a threshold, related to the crystal lattice energy of copper.

The effects of gas hourly space velocity, temperature and pressure of reaction gases on the catalytic performance of 40Cu under dynamic activation are shown in Figs. 3b–2d. The rate of $CO_2$ conversion at GHSV = 21600 ml·g$_{cat}^{-1}$·h$^{-1}$ is about three times that at GHSV = 9600 ml·g$_{cat}^{-1}$·h$^{-1}$. This is an anomalous phenomenon, not in line with common rules of catalysis. At temperatures lower than 300 °C, CO formation is almost completely inhibited, and methanol becomes the main product in a selectivity of ~95%. It suggests the catalyst at higher GHSV has more sites and/or sites more active for methanol formation, and CO desorption is difficult from the surface under dynamic activation. Higher temperature (350 °C) can release the inhibition on CO desorption from dynamic activation catalyst 40% Cu/Al$_2$O$_3$ (Fig. 3c). As to the pressure effect (Fig. 3d), the increase in pressure is remarkably beneficial to the conversion of $CO_2$ and the formation of methanol, while further limiting the CO formation.

For the catalyst 20Cu, as shown in Fig. 3e, however, there is hardly any significant difference in the catalytic performance for $CO_2$ hydrogenation, whatever the mode of DAR or FBR used, due to the stripping energy required for the 1–2 atomic layers of Cu is large, originated from the strong interaction between the copper and alumina (Supplementary Table 2), and the impact force has little effect on the catalyst surface.

Figure 3f shows the activation energy of $CO_2$ conversion for the reaction under different conditions. The apparent activation energy for $CO_2$ conversion over 40Cu (DAR) is significantly lower than that in the traditional or static mode of reaction (FBR). On the contrary, the 20 Cu shows a much larger apparent activation energy of $CO_2$ conversion, both under DAR and FBR. The activation energies are consistent with the extremely large leap in performance of catalysts in different modes of reaction and indicate that the dynamic activation of 40 Cu is far distinct from its static state in fixed-bed reaction.

Figure 3g depicts the results of the reaction switched from the space velocity of 21,600 to 9600 ml·g$_{cat}^{-1}$·h$^{-1}$, corresponding to gas flow rate of 360 to 160 ml·min$^{-1}$ with 1 g catalyst loaded. Firstly, the steady-state reaction was carried out for three hours at 21,600 ml·g$_{cat}^{-1}$·h$^{-1}$ and then, the space velocity was switched to the reaction at 9600 ml·g$_{cat}^{-1}$·h$^{-1}$. It can be seen from the figure that the reaction performance gradually deteriorated to the steady-state reaction at GHSV 9600 ml·g$_{cat}^{-1}$·h$^{-1}$, in seven hours. Considering the time needed for the replacement of the dead volume of the reactor (Supplementary Fig. 2), the relaxation time of dynamic activation state to the static state of the catalyst is about 2 h, a bit unexpected long. This result indicates that the active sites generated under dynamic activation are metastable in thermodynamics and will relax back to the static state in the absence of a continuous collision energy supply, but the relaxation is not so rapid.

The effect of hotspots induced by collisions can be excluded from the current investigation, at least, it is not the main factor. The catalytic performance of the same catalyst was tested in Fixed-bed reactor (FBR) at higher temperatures (Supplementary Fig. 8), and it showed increased $CO_2$ conversion but >~90% CO selectivity, contrasting DAR's ~95% methanol selectivity and 660 mg·g$_{cat}^{-1}$·h$^{-1}$ methanol space-time yield. This aligns with mechanochemical literature[23], which suggests negligible localized heating in certain mechanically driven systems. In addition, the same catalyst was also tested using a stirred ball mill reactor (SBMR) with ~90 W stirring power for $CO_2$ hydrogenation, giving a methanol space-time yield of 230 mg·g$_{cat}^{-1}$·h$^{-1}$ and CO selectivity as high as ~75%, but the catalyst deactivated within 120 minutes and the SBMR induced severe lattice collapse of 40Cu (Supplementary Fig. 9).

To validate the above changes in copper states, the samples before and after the reaction were fully characterized. Figure 4a depicts the XRD patterns of the 40Cu catalyst after 2 h of reaction under dynamic activation. Compared to fresh 40Cu (Fig. 4b), the XRD peaks of Cu after 2 h on stream shift to smaller 2θ angles and become broader and less intense (Fig. 4c), revealing that the Cu crystallinity decreases and tends to be more discrete[24]. The lattice parameters, cell volume, microstresses, and bond lengths of Cu have increased to a certain extent by the analysis using Rietveld refinement (Fig. 4a, b, Supplementary Table 2), which means that the stresses in interior of the metal particles generated by the collision cause the crystal of Cu distorted, less crystalline, Cu-Cu distance elongated and the coordination number of the Cu atoms decreased[25], referenced to the radial distribution functions shown in Fig. 1.

To determine the changes in the coordination environment of the Cu catalyst under dynamic activation, its extended X-ray absorption fine structure (EXAFS) spectra are collected to investigate the local structure of Cu. The results of the Fourier transform of the Cu K-edge EXAFS, shown in Supplementary Fig. 10, indicate the Cu-Cu scattering paths[26]. By further adopting the nonlinear least squares fitting analysis, the coordination number of Cu-Cu for the sample under dynamic activation is clearly reduced (9.7 ± 0.7 → 7.8 ± 0.7) and the bond length of Cu-Cu has been elongated to a certain extent (2.535 Å→2.539 Å, Supplementary Table 3), which is consistent with the results of XRD Rietveld refinement. Moreover, our samples—relaxed for more than 2 h post-reaction—further narrow the gap between post-reaction and fresh catalysts, yet still facilitate analysis of impact-induced Cu-Cu structural shifts. In contrast, no significant changes can be identified for the catalyst tested in a traditional fixed bed, in terms of the coordination number and bond length of Cu-Cu before and after the reaction.

As further shown by aberration-corrected HAADF-STEM (Fig. 4d), the 0.211 ± 0.001 nm of Cu (111) lattice fringe distance for 40Cu under dynamic activation is larger than that of fresh 40Cu (0.208 ± 0.001 nm) and the halo ring featured amorphous phase of Cu to some extent is distinguishable, indicating that even after a long period of standing time at room temperature for relaxation after reaction, the effect of dynamic activation on the crystalline Cu can still be detected clearly. However, there is no significant change in Cu (111) of 40Cu (0.208 ± 0.001 nm) after the static reaction in the traditional fixed-bed (Supplementary Figs. 11, 12). This reveals that Cu partially loses its lattice features under dynamic activation. In addition, the aberration-corrected HAADF-STEM image and the corresponding EDS elemental mappings show that significant migratory agglomeration occurs in 40Cu during the static reaction in traditional fixed bed, and the average size of the Cu nanoparticles increases from 4.04 nm to 6.81 nm (Supplementary Fig. 12), which corresponds to the poor stability of the catalysts in the static reaction as described above. For the catalyst after 24 h of reaction under dynamic activation, however, the Cu nanoparticles reduced their average size to 3.08 nm, showing significantly increased stability.

## Theoretical analysis

With the dynamic activation, the continuous import of reaction gas ensures that the collision between the Cu catalyst and the rigid target occurs uninterruptedly to create highly active sites usually unavailable. About the exact surface state of the catalyst during operation, we performed molecular dynamics simulations to describe the collision process and captured snapshots of the instantaneous state of the catalyst for analysis using DFT calculation (Fig. 5). The distribution of gas flow rate in the dynamic activation reactor was firstly calculated by Fluent software using a discrete phase particle model simulation and shown in Supplementary Fig. 1. The coordination number of Cu in different snapshots was obtained by integrating the radial distribution function of Cu and the average Cu-Cu bond length was obtained by averaging the spacing of the first coordination shell[27]. Consistent with

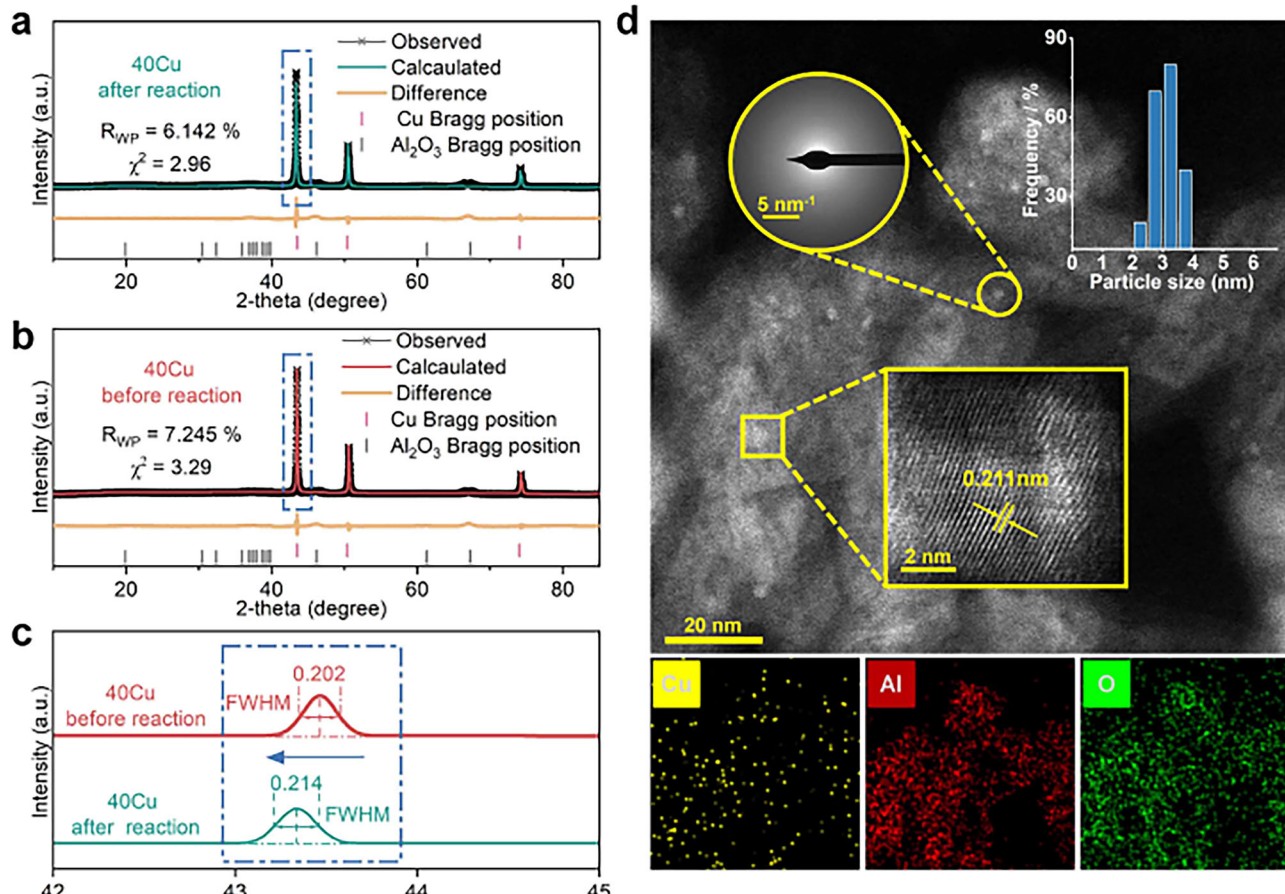

**Fig. 4 | Characterization of catalysts about structural variation upon collisions and reaction.** Rietveld refinement of XRD patterns of catalyst 40Cu. **a** After 2 h on stream of activated catalytic reaction; **b** Before reaction. **c** Local enlargement between 2 theta 42–45°. **d** Aberration-corrected HADDF-STEM images and elemental mappings of 40Cu after reaction of $CO_2$ hydrogenation under dynamic activation.

intuition, a large number of coordination unsaturated sites are exposed on the surface (Supplementary Fig. 13) and the average coordination number obtained is 7.02, close to the results of EXAFS (8.4 ± 0.8) but significantly less than that of the pristine copper (12). The bond length is significantly increased to 2.69 Å, much larger than that of normal Cu-Cu bond (2.535 Å). The copper catalyst exhibits an unusual state which we defined as discrete condensation. It generally retains the crystalline form of Cu, but has discrete tendency and certain amorphous.

As to the collision energy, in the dynamic activation reactor (DAR), an input power of ~0.34 W by reaction gas flow pro Macterized by lattice distortion and markedly enhances catalytic performance. Density functional theory (DFT) calculations confirm that the discrete condensed Cu state has an energy of -0.17 eV/atom higher than the normal state of Cu, validating this structural transition.

According to the above two states of the catalyst, discrete condensed state and pristine state on alumina, we further performed DFT calculations to know the reaction mechanism related to $CO_2$ hydrogenation over 40Cu in different states. Figure 5a, b depict models and bond length of the two states of Cu. Based on the elementary steps optimized, the calculated energy profiles for methanol and CO formation on both models are shown in Fig. 5c, e and the structures of the initial, transition and final states for each elementary step are given in Fig. 5d, f respectively.

As is well known that the whole reaction initiates with $CO_2$ activation. It is found that the adsorption energy of $CO_2$ on 40Cu-pristine and 40Cu-discrete condensed is +0.28 and -0.19 eV, respectively. In

addition, it is noteworthy that the activation barrier for the adsorbed $CO_2^*$ hydrogenated to $COOH^*$ is slightly lower on the discrete condensed one. These results indicate that the supported copper in the discrete condensation state is more beneficial to $CO_2$ adsorption and activation. Since the RWGS reaction is a competitive reaction to methanol synthesis during the process of $CO_2$ hydrogenation, the fate of those generated $CO^*$ species from $COOH^*$ is pivotal to the product distribution[28]. On the 40Cu-pristine surface (Fig. 5c), the energy barrier for the further hydrogenation of $CO^*$ was 1.15 eV, which was a little bit higher than the energy required for $CO^*$ desorption from the surface (0.99 eV). This implies the RWGS reaction would be dominant in the hydrogenation process. However, this is totally different for 40Cu-discrete condensed (Fig. 5e). In contrast with the desorption of $CO^*$ (ΔE = 1.13 eV), subsequent hydrogenation into $HCO^*$ (Ea = 0.74 eV) is much easier. More interestingly, further hydrogenation of $HCO^*$ to $H_2CO^*$, $H_3CO^*$, and $CH_3OH^*$ on the surface of 40Cu-percussive surface is also more favorable in terms of both thermodynamic and kinetic levels. The above analyses are well consistent with the former experimental results. Overall, the metastable 40Cu in the discrete condensed state would remarkably inhibit the formation of the CO by-product, thus leading to the super high selectivity of methanol.

From a practical perspective, the future of the dynamic activation reactor (DAR) would be highly important. Essentially, the DAR is also a fluidized-bed reactor with a special collision and reaction zone. Based on present results over the common catalyst of alumina-loaded copper, in concepts for the production of a 10,000-ton/year methanol, only ~1.8 tons of catalyst would suffice. Structurally, the reactor is in

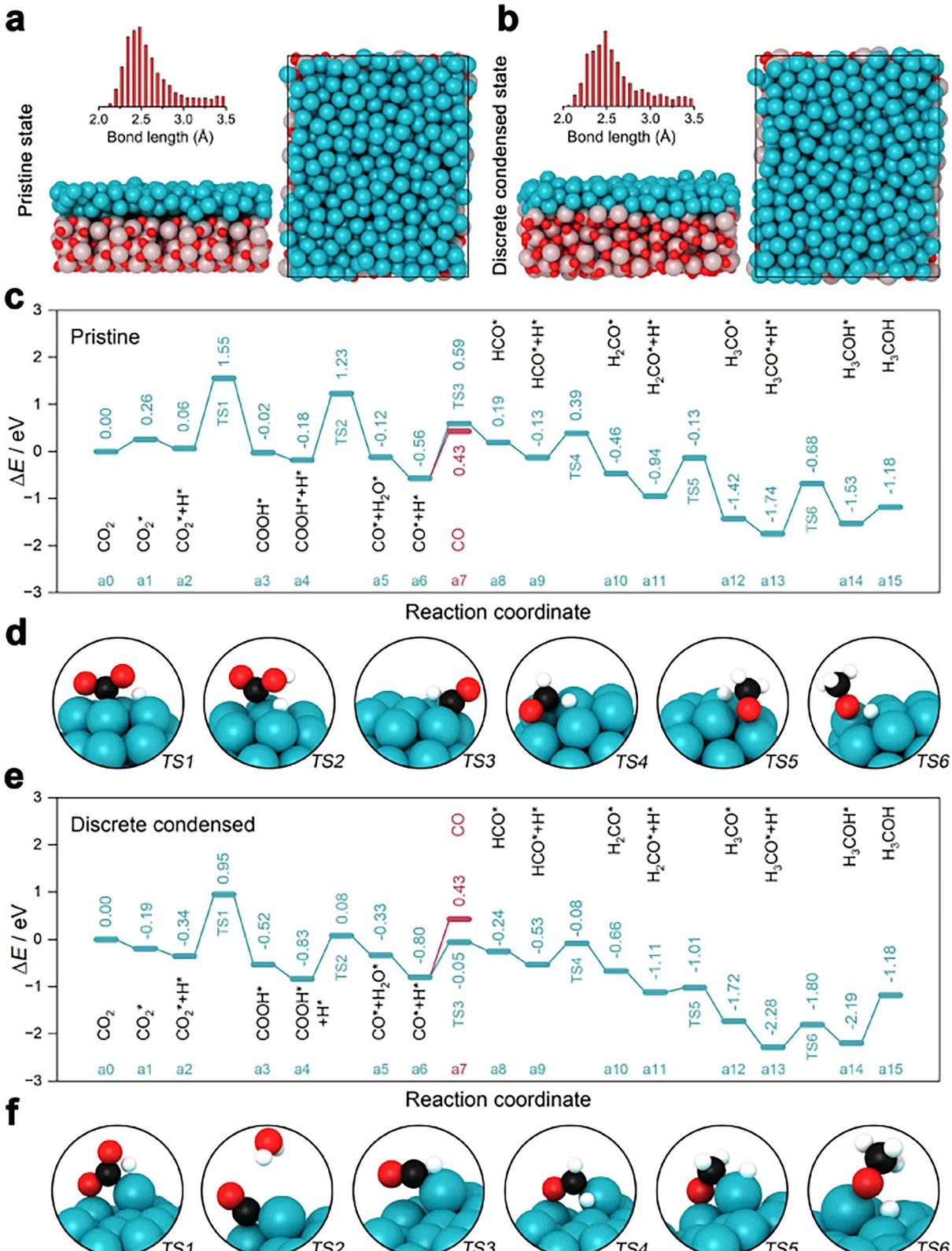

**Fig. 5 | Explanation and understanding by calculation chemistry.** Top and side views of snapshots of molecular dynamics simulations and the Cu-Cu distance distribution of 40Cu in different states. **a** Pristine state of copper; **b** Dynamic activated state of copper; (red, white, and cyan balls denoting O, Al and Cu atoms, respectively). Energy profiles of $CO_2$ hydrogenation over 40Cu. **c** Pristine state of copper in a traditional fixed-bed reactor. **d** Configurations of the initial, transition, and final states involved in methanol synthesis over the model of pristine $Cu/Al_2O_3$. **e** Discrete condensed state of copper with dynamic activation. **f** Configurations of the initial, transition, and final states involved in methanol synthesis over the model of discrete condensed $Cu/Al_2O_3$.

fact a combination of a gas flow mill and a cyclone separator. Through structural optimization design, its volume can definitely be designed very small. More innovative and usable would be expected from this catalytic reaction mode.

## Discussion

Dynamic activation catalysts driven by the kinetic energy of the reaction gas itself have been first demonstrated, in which catalyst particles, blown by reactive gas flow, impact a rigid target uninterruptedly at high speed to create the dynamic activation state, with more sites and/or sites more active. The dynamic activation 40% $Cu/Al_2O_3$ shows a three times enhanced rate of $CO_2$ conversion, with methanol selectivity increased to 95% from less than 40%. The space-time-yield of methanol is six times enhanced to 660 mg·g$^{-1}$·h$^{-1}$. Combined results of experimental and theoretical investigation, the dynamic activation $Cu/Al_2O_3$ is defined as a discrete condensed state, energetic and metastable, never described before, possessing distorted and less crystalline copper, elongated Cu-Cu distance, and reduced coordination number. The dynamic activation catalysts are constantly changing at every moment, interpreting another mechanism of stability. The method can create unusual catalysis that cannot be observed in traditional catalytic systems, and continuous work will enable us to regulate catalysts in a much broader scope and discover more advanced catalytic reactions.

## Methods

### Chemical reagents

Cu $(NO_3)_2$ 3$H_2O$ was purchased from Aladdin Reagent Co., Ltd. A new type of γ-$Al_2O_3$ material with a high-energy surface as the main external surface was produced according to the preparation method previously reported by our group[29].

### Catalyst Preparation

Synthesis of x$Cu/Al_2O_3$. Firstly, the precursor Cu $(NO_3)_2$·3$H_2O$ was added to a beaker (100 ml) with 20 ml of water and a stirring magnet for stirring for 5 min, after which support powder ($Al_2O_3$), corresponding to the catalyst prepared, was added to continue stirring. Then the beaker was fixed inside a water bath to continue stirring, and the water temperature was 80 °C. After stirring for about 5 h, until the water evaporated completely, the catalyst was dried at 110 °C for 12 h and calcined at 450 °C for 3 h (10/min) in flowing Argon. Catalyst x$CuO/Al_2O_3$, was obtained with x corresponding to the percentage copper content in the catalyst. Before the catalyst test, the catalyst was reduced to metallic copper in the reactor. The catalyst is placed in a reactor and calcined at 300 °C for 3 hours under a $H_2/N_2$ atmosphere with a ratio of 3:1, followed by cooling to room temperature.

### Characterizations

X-ray diffraction (XRD) patterns of the powders were recorded in the 2θ range from 5 to 90° using Cu Kα radiation (40 kV, 40 mA) on an X'pert PAN analytical diffractometer. The Brunner-Emmett-Taylor (BET) specific surfaces were analyzed using a Micromeritics ASAP2420 analyser at a temperature of liquid nitrogen (77 K), with the samples degassed at 300 °C for 4 hours under a vacuum of 10$^{-3}$ torr prior to testing. The K-edge (8.979 keV) X-ray absorption fine structure (XAFS) analysis of copper was carried out at the Shanghai Synchrotron Radiation Centre, beamline BL14W1 (Shanghai, China). The EXAFS spectra were processed and analyzed by the software Athena, and the corresponding fitted data were obtained by the software Artemis. HAADF-STEM images of x$Cu/Al_2O_3$ were obtained on an FEI TalosF200S with an accelerating voltage of 200 kV, and the elemental distributions of O, Al, Cu, and. Spherical aberration-corrected scanning transmission electron microscopy (STEM) was performed on a Titan Themis G2 with an accelerating voltage of 300 kV using the integrated differential phase contrast (iDPC) detection mode.

### Catalytic test

The $CO_2$ hydrogenation reaction is performed on a dynamic activation reactor (DAR). In a typical procedure, 1 g of sample powder was packed inside the reaction. Initially, the air in the reactor was pumped out using a vacuum pump before heating. The reaction gas was then introduced into the reactor at the flow rate needed.

For comparison, the catalytic property of the normal state of $Cu/Al_2O_3$ was tested in a fixed-bed reactor. The reactor tube is in a double-layer structure, the inner and outer tube is made of quartz and stainless steel, respectively. In a typical procedure, 0.2 g of sample pellets (20–40 mesh) is packed into the quartz tube with an inner diameter of 0.6 cm. The reaction gas flow rate was set at needed to give the same GHSV as that in the dynamic activation reactor.

The reaction tail gases were fed into a gas chromatograph (GC-9860) equipped with a flame ionization detector (FID) and a thermal conductivity detector (TCD) for online analysis. A Plot Q capillary was connected in front of the FID detector and TDX-1 packed column to detect organic products and $CO_2$, Ar, and CO gases in the tail gas using TCD, respectively. All substances were determined qualitatively by comparing retention times and standards. The catalytic data were calculated using argon (Ar) as an internal standard, and the corresponding methods were as follows:

$$CO_2 \text{conversion} = \frac{f_{CO_2^{in}} \cdot A_{CO_2^{in}} - f_{CO_2^{out}} \cdot A_{CO_2^{out}}}{f_{CO_2^{in}} \cdot A_{CO_2^{in}}} \times 100\% \quad (1)$$

$$CO \text{ selectivity} = \frac{f_{CO^{out}} \cdot A_{CO^{out}}}{f_{CO_2^{in}} \cdot A_{CO_2^{in}} - f_{CO_2^{out}} \cdot A_{CO_2^{out}}} \times 100\% \quad (2)$$

$$CH_3OH \text{ selectivity} = \frac{f_{CH_3OH} \cdot A_{CH_3OH}}{f_{CH_3OH} \cdot A_{CH_3OH} + f_{CH_4} \cdot A_{CH_4}} \times 100\% \quad (3)$$

$$CH_4 \text{ selectivity} = \frac{f_{CH_4} \cdot A_{CH_4}}{f_{CH_3OH} \cdot A_{CH_3OH} + f_{CH_4} \cdot A_{CH_4}} \times 100\% \quad (4)$$

$$R_{CH_3OH} = \frac{CO_2^{in} \cdot X_{CO_2} \cdot S_{CH_3OH} \cdot M_{CH_3OH}}{22.4 \cdot m_{cat}} \times 100\% \quad (5)$$

$$R_{CO} = \frac{CO_2^{in} \cdot X_{CO_2} \cdot S_{CO} \cdot M_{CO}}{22.4 \cdot m_{cat}} \times 100\% \quad (6)$$

where $CO_2^{in}$, $CO^{in}$ and $CO_2^{out}$, $CO^{out}$ represented mass of $CO_2$ and CO at the inlet and outlet, respectively.

The input energy is evaluated using the following equation:

$$P_K(W) = \frac{1}{2}mV^2 \quad (7)$$

where $P_K$ is the Kinetic power, measured in watts (W, or J/s), m is the mass of the Gas in kilograms (kg), and V is the Gas velocity, in meters per second (m/s).

## Data availability

All data are available upon request to the corresponding author.

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

## Acknowledgements

This work was supported by the National Natural Science Foundation of China (21932004, 91963206, 22408151) and the Ministry of Science and Technology of China (2021YFA1500301). The support from the NJU-HUACHANG Joint Institute of Meso Catalysis was also appreciated.

## Author contributions

WD: conceptual design, supervision, method design, review, and editing. Z.Z.: method design, drafting, research implementation, and visualization. JY: computational research, method design. C.S.: computational research, visualization, method design. F.L., C.D., T.Z., X.G., Y.Z., X.K.G., N.X., and L.P. made contributions to this paper through frequent discussions.

## Competing interests

The authors declare no competing interests.
