## [Transparent Peer Review file · Nature Communications]

Dynamic Activation Catalysts for CO₂ Hydrogenation

Corresponding Author: Professor Weiping Ding

A version of this paper was originally rejected for publication by Nature Communications, however that decision was reconsidered after appeal by the authors.

Version 0:

Reviewer comments:

Reviewer #1

(Remarks to the Author)

The reactor design outlined in this work seems like a very complex mechanochemistry system, which is interesting, but I have significant doubts about the claims made in this work.

1. There is no discussion or even reference to mechanochemistry (Tricker, Andrew W., et al. "Hot spot generation, reactivity, and decay in mechanochemical reactors." *Chemical Engineering Journal* 382 (2020): 122954) which operates under similar principles as this reactor.

2. there is no discussion of localised heating of the Cu catalyst particles when they collide with the wall of the stainless steel reactor.

3. The authors claim "the coordination number of Cu-Cu for the sample under dynamic activation is clearly reduced ($9.4 \pm 0.9 \rightarrow 8.4 \pm 0.8$) and the bond length of Cu-Cu has been elongated to a certain extent ($2.535 \text{ \AA} \rightarrow 2.539 \text{ \AA}$, table S4)". In the first measurement, the errors overlap. In the second measurement a change in 0.004 \AA is not resolvable from XAS fitting. Also, this difference in bond length is so small it would be dwarfed by any vibrational motion.

4. Similarly, the authors claim "As further shown by aberration-corrected HAADF-STEM (Fig. 3d), the 0.211 nm of Cu (111) lattice fringe distance for 40Cu under dynamic activation is larger than that of fresh 40Cu (0.208 nm ". I do not know of any TEM that can resolve features at 0.03 \AA in length.

My assumption based on the results shown in this paper is that by increasing the gas flow rate they are colliding with the target with higher kinetic energy, which locally heats the catalyst particles, increasing the turnover rates. I feel that this paper is not suitable for publication, and needs to be completely re-worked.

Reviewer #2

(Remarks to the Author)

This paper claims that accelerating a catalyst powder in a stream of CO₂/H₂ at a stainless steel target leads to an improvement in the catalytic reactivity by a factor of 3. They claim that selectivity to generate CH₃OH is more than doubled. They claim that this is due to the velocity of the catalyst being 75 m/s impacting the stainless steel. This work is based on a Cu/Al₂O₃ catalyst for the first part.

The particles are in a flow of up to 75 m/s and then impact a stainless steel target and they are shaken up every 3 s by tapping. They need to show the apparatus in the paper, not in the SI and provide more visual details about the experiment in the paper on the apparatus. They also need to add a timing diagram so we can see when the flow is present and when it is turned off. Is the flow maintained, or do they shut off the flow after the initial impact? Is this what is shown in figure 2g?

Remove lines 28 to 32 as they are not relevant.

In the center of mass how much is the velocity changed for particles and the gas? With what energy do the catalyst particles hit the surface and how much energy is there to change the properties of the catalyst particles? How much kinetic energy is there on the impact with the stainless steel? Is this enough to change the surface structures of the catalyst?

I think what they are showing is that the collision with surface changes the morphology. This concept is not new. The way to do it may be new, but the consequences are not new or novel. As this is essentially a mechanical force applied to the surface to distort it, how much force is there is other means of doing this are present. How dependent are the catalyst changes on the flow velocity and hence the impact energy?

I think that much of what they have done may also be to get more surface area available for the catalyst. How much more surface area is there? Also, as the particles are in the gas phase due to collisions and tapping, how does that affect the surface area?

How much of what is happening is just forming more active sites by collision with the surface and any reactions with what is on the steel surface to change particle morphology?

This is not really a new method of dynamic activation in my opinion. They are just able to get more surface area and potentially more active sites by colliding the particles with the surface and then getting the particles back into the gas phase.

What is the effective temperature of the system on collision?

It seems that this is just mechanical changes to the catalyst surfaces and this is not novel.

They need to move Table S2 to the paper.

Line 91 'latter' not 'later'.

The authors should look at the fast reactor flow work of Lanny Schmidt from the University of Minnesota.

Error bars are needed for all of Figure 2.

How reproducible is the work?

Line 144, remove 'elaborate' as there is nothing special in terms of the characterization. They could change it to extensive

Line 193. At least state the type of DFT calculations.

Much more details are needed to reproduce the Fluent results. What is given is not enough. There is no information meshing, what parts of Fluent were used, etc.

How was the frequency analysis done for the transition state DFT calculations?

What type of energy for figure 4? Electronic energy? Free energy?

How good is the force field used for the MD simulations to get the coordination at Cu correct as well as the bond distances? What does the MD really show? How good are the MD energetics?

In the MD simulations, how much higher in energy is the distorted structure than the non-distorted one? How much higher at the DFT level?

What are two states of the Cu in the DFT simulations? Were the complete DFT structures optimized? If so, why did they not relax to the lowest energy one? How big is the unit cell?

The work on the MoS₂ is just tacked on and there are insufficient details to reproduce it or to gain anything from what is there. Remove it and focus on the CuO on Al₂O₃.

line 246. What is STY?

Can this process be used for at an industrial level? What is the energy input for the flow and the tapping?

The entire paper is too terse and there is no discussion of the accuracy of any of the experiments or calculations.

I am not sure there is anything new or novel here except that mechanical means can distort a catalyst making more active sites and that particles in the gas phase may have more surface area.

Version 1:

Reviewer comments:

Reviewer #2

(Remarks to the Author)

The manuscript and SI are substantially improved. I am not sure that I agree with all of the authors' conclusions, but I think that publication is warranted so the research community can see this work and comment on it or critique it.

I have one additional request. The authors provide excellent information in the response to the review. Please make sure that all of the responses are at least in the SI. include responses to comment 2-3, comment 2-7, and comment 2-18.

They should also provide some comment on the potential for practical use of this approach at industrial scale. See comment 2-23.

This would involve very minor revision.

Reviewer #3

(Remarks to the Author)

Ding et al. report on a highly unusual reactor set-up, which uses a rapid stream of CO₂/H₂ to mechanically mix Cu/Al₂O₃ catalyst particles, resulting in greatly increased activity and selectivity for methanol, compared to conventional thermocatalysis. They provide an explanation in the form of mechanochemical activation of the Cu particles, caused by impacting on the reactor backstop at high velocity, resulting in altered changed Cu nanoparticle structure. However, they are unable to convincingly demonstrate this changed structure, relying on statistically insignificant differences in XRD and TEM data. Computational explorations are provided, however these simulations amplify the alleged differences between the activated and unactivated catalyst structures, reducing the reliability of these results. Previous criticisms of insufficient detail to reproduce results have been adequately dealt with (to the best of my knowledge). The authors have certainly discovered an interesting phenomenon, but have not been able to convincingly explain it.

1. It seems unlikely to me that the velocity numbers added to the abstract would be well understood by readers in the field of heterogeneous catalysis.

2. Line 36: "to precisely activate the catalyst surface" – I don't believe there is anything chemically "precise" about this reactor (or mechanism of activation), and would suggest an alternative wording.

3. Line 43: Djéga-Mariadassou and Boudart, 2003 (Reference 7) state: "The QSSA theory is based on three fundamental rules: (i) concentrations of intermediates are very low; (ii) the variation of the concentration of one intermediate is stationary, i.e., independent of time ($d[\text{intermediate}]/dt=0$); (iii) consequently, rates of all steps in a sequence have the same value, once divided by the stoichiometric number σ_i , the number of times we need an i -elementary step in the sequence to obtain the overall chemical equation." With respect to the line in this work: "challenging the traditional notion about the relationship of reaction rate with space velocity", I would content that when the assumptions of Djéga-Mariadassou and Boudart are sufficiently perturbed, new behaviour would not be surprising, but expected. Could the authors comment further on the differences between their system and the 'traditional' kinetic analysis, and specifically which assumptions need to be reevaluated?

4. Fig. 2 is too complex to readily absorb the information on the reactor design. Fig. 2 should be reproduced in clear, schematic form, with the reactor components properly sized and labelled. For example, there is no need to represent a GC in this figure.

5. What is the catalyst particle size? How does this change after the reaction? How are the catalyst particles prevented from being blown through the reactor?

6. The STYs in Figs. 3b, 3c and 3d should be in mmol gcat⁻¹ h⁻¹, so that the species can be compared directly.

7. Line 186: "the coordination number of Cu-Cu for the sample under dynamic activation is clearly reduced (9.4±0.9→8.4±0.8)" – these numbers are the same, within error. The supposed Cu-Cu bond elongation (2.535 Å→2.539 Å) is also remarkably small – these numbers require an associated error (errors are given in Table R1 which suggest that the distances are identical). The difference appears to be far below the resolution limit of XRD (which is closer to 1 Å than 0.01 Å, depending on the X-ray frequency).

8. Similarly, the lattice fringe distance has a quoted error of only 0.001 nm (i.e. 0.01 Å), which is unbelievably small

9. Of concern is the discrepancy between the Cu-Cu distances and the CNs in the text vs in Table R1 (particularly the much lower CN in the table than in the text). Which of these is the correct value?

10. Previous Reviewer's Comment 2-4 is not sufficiently engaged with – I agree with the previous reviewer that "How dependent are the catalyst changes on the flow velocity and hence the impact energy?" is a key question that remains unanswered.

11. Previous Reviewer's Comment 2-12 has not been adequately answered – how have the error bars in the new Fig. 3 been calculated?

Version 2:

Reviewer comments:

Reviewer #3

(Remarks to the Author)

The authors have engaged very thoroughly with the comments from this review round, further improving the manuscript. The authors have also brought more evidence (or revisited the existing data) to support some of their more surprising findings. I recommend publishing for the benefit of the wider catalysis community.

Response to the reviewers' comments

We really thank the reviewers for their instructive comments. We have made revision of the manuscript, considering these comments and suggestions. All the changes have been highlighted in yellow background in the revised manuscript and supporting information. The point-to-point responses to the reviewers' question are attached bellow.

T

Reviewer 1: The reactor design outlined in this work seems like a very complex mechanochemistry system, which is interesting, but I have significant doubts about the claims made in this work.

Response: In fact, this reactor is not complicated at all, there are no any mechanical moving parts.

Comment 1-1: There is no discussion or even reference to mechanochemistry (Tricker, Andrew W., et al. "Hot spot generation, reactivity, and decay in mechanochemical reactors." *Chemical Engineering Journal* 382 (2020): 122954) which operates under similar principles as this reactor.

Response: Indeed, both the dynamic activation reaction (DAR) reported by us in this work and the mechanochemistry reported in references harnesses mechanical force to drive reactions, but their way of working markedly differs from each other. By the dynamic activation reaction (DAR), we utilize the kinetic energy of the reaction gas itself to carry catalyst particulates to collide cyclically with a rigid target, inducing catalyst surface changes that significantly enhance their catalytic properties such as CO₂ hydrogenation to methanol (Figure 3). The energy imported is basically precise and homogeneous and the upper limit of kinetic energy carried by the reaction gas is ~0.34 joule per second (corresponding power is ~0.34 Watt), acting on one gram of catalyst. In contrast, the energy utilized by mechanochemistry, such as ball milling, is much higher and relies on solid-state grinding, generating localized high-temperature hotspots (Chem. Eng. J., 2020, 122954.), primarily for solid-phase reactions. In this report, we highlight the dynamic activation reaction (DAR) just for gas-solid catalytic reactions, not ignoring the contribution of mechanochemical research.

To address this, we have performed similar experiments for CO₂ hydrogenation in a stirred ball milling reactor (SBMR, silica balls), with a stirring power of ~90 W, and the results are shown in Figure R1 below. The methanol space-time yield (STY) over the same 40Cu catalyst was ~230 mg·g_{cat}⁻¹·h⁻¹, with methanol selectivity less than 25% and CO selectivity higher than 75%, which

significantly deactivated within 120 minutes (Figure R1a, b). The catalyst 40Cu experienced severe lattice collapse after reaction (Figure R1c). DAR's lower input power (~0.34W from nozzle jet kinetic energy) effectively induces morphological changes while maintaining catalyst stability. This suggests that the catalyst's surface properties underwent severe alterations under SBMR conditions, exacerbating the occurrence of CO side reactions. Relatively, the energy impacted on a catalyst particle in DAR reactor is more predictable, for example, no extra activation occurs at gas flow of 160 ml·min⁻¹(Fig. S6), whereas 360 ml·min⁻¹ boosts STY of methanol nearly threefold, underscoring impact energy's critical role. The DAR uniquely uses gas flow to regulate the reaction.

The input energy is evaluated using the following equation:

$$P_K (J/s) = \frac{1}{2}mv^2$$

where P_k is the Kinetic power, measured in watts (W, or J/s), m is the mass of the Gas in kilograms (kg), and v is the Gas velocity, in meters per second (m/s).

In the revised manuscript, some discussions about mechanochemistry have been added (lines 151~).

Figure R1. (a) Catalytic performances of 40Cu in stirred ball mill reactor (SBMR). (b) Formation rates of CO and methanol over 40Cu measured in stirred ball mill reactor (SBMR) and dynamic activation reactor (DAR). (c) Blank test of CO₂ hydrogenation measured using the stirred ball mill reactor (SBMR). (d) Rietveld refinement of XRD patterns of catalyst 40Cu. (P: 2.0 MPa; T: 300 °C; Cat: 0.2 g; Flow rate: 72 ml/min, if unspecified).

Comment 1-2: there is no discussion of localised heating of the Cu catalyst particles when they collide with the wall of the stainless steel reactor.

Response: This is good question, but quantitative estimation is extremely difficult. In current dynamic activation reaction (DAR), catalyst particles (40Cu) are driven by high-velocity reaction gas stream to collide cyclically with a target, inducing mechanical strain that alters their structure.

Considering the energy concerned, ~ 0.34 J/s impacted on 1 g catalyst in the high-velocity reaction gas stream with heat dissipation, the hotspot is much weak, compared with that in ball milling reactor.

Figure R2. (a), (b) Catalytic performance of 40Cu under different temperature conditions in traditional fixed bed. (P: 2.0 MPa; Cat: 0.2 g; Flow rate: 72 ml/min).

To address this, a control experiment using a fixed-bed reactor (FBR) with 40Cu at varying temperatures was performed. As shown in Figure R2, CO₂ conversion rate increases with the temperature, but the product was almost exclusively the byproduct CO, totally different from the DAR reactor, as least reflecting the temperature is not a key factor, even localized. Some unique active sites induced by collision of catalyst in DAR with uniform and controllable energy is the scientific essence of DAR, that is the reason we called it as dynamic activation reaction.

Discussion about local heating effect has been added in the revised manuscript to provide a more comprehensive explanation of the dynamic activation mechanism (cf. lines 151~).

Comment 1-3: The authors claim "the coordination number of Cu-Cu for the sample under dynamic activation is clearly reduced ($9.4 \pm 0.9 \rightarrow 8.4 \pm 0.8$) and the bond length of Cu-Cu has been elongated to a certain extent ($2.535 \text{ \AA} \rightarrow 2.539 \text{ \AA}$, table S4)". In the first measurement, the errors overlap. In the second measurement a change in 0.004 \AA is not resolvable from XAS fitting. Also, this difference in bond length is so small it would be dwarfed by any vibrational motion.

Response: Thanks for your careful reading. We cannot measure the atomic coordination number and bond length of the catalyst under dynamic conditions. After the reaction is completed, the catalyst is basically restored to the state before reaction through cooling and structural relaxation. Nevertheless, we definitely measured the traces left by the dynamic changes of the catalyst, as

shown in the revised Fig. 3g of the manuscript.

To ensure the reliability of the EXAFS analysis, we re-analyzed the data and listed the results in Table R1. In the initial fitting, only the amplitude reduction factor (S_0^2) of Cu foil was used as a reference, with S_0^2 determined independently for each sample. In the revised fitting, we standardized the approach by fixing the coordination number of Cu foil at 12 to determine S_0^2 , which was then held constant across the fitting of three sample sets (dynamically activated, static reaction, and fresh catalyst), enabling precise quantification of coordination number (CN) changes. The final fitting results satisfy the following constraints: energy shift $\Delta E_0 < 10$ eV, first-shell Debye-Waller factor $\sigma^2 < 0.01 \text{ \AA}^2$, and bond length variation $\Delta R < 0.1 \text{ \AA}$, ensuring the robustness and validity of the data. These refinements confirm that the structural changes induced by dynamic activation are statistically significant, consistent with the lattice distortions observed via XRD and HAADF-STEM (Fig. 4). We have corrected this in the supporting information.

Table R1. Cu K-edge EXAFS data for the catalysts ^a.

Samples	Shell	R (\AA) ^b	CN ^c	σ^2 (\AA^2) ^d	ΔE_0 (eV)	r-factor
40Cu-fresh	Cu-Cu	2.537±0.005	9.7±0.7	0.0075	2.27	0.009
40Cu (FBR)-used	Cu-Cu	2.535±0.006	9.8±0.8	0.0079	2.05	0.011
40Cu (DAR)-used	Cu-Cu	2.538±0.005	7.8±0.7	0.0085	2.41	0.009
Cu foil	Cu-Cu	2.537±0.007	12*	0.009	5.18	0.012

a: The S_0^2 value are 0.869 according to the experimental EXAFS fit of Cu foil by fixing CN as the known crystallographic value for Cu K-edge EXAFS spectra fitting, respectively; b: bond length; c: coordination number; d: Debye-Waller factor.

Comment 1-4: Similarly, the authors claim "As further shown by aberration-corrected HAADF-STEM (Fig. 3d), the 0.211 nm of Cu (111) lattice fringe distance for 40Cu under dynamic activation is larger than that of fresh 40Cu (0.208 nm)". I do not know of any TEM that can resolve features at 0.03 \AA in length.

Response: Thanks for the sharp question. Refer to the previous question, we cannot measure the atomic coordination number and bond length of the catalyst under dynamic conditions. After the reaction is completed, the catalyst is basically restored to the state before reaction through cooling

and structural relaxation. Modern aberration-corrected HAADF-STEM instruments achieve point resolutions below 0.05 nm, and with advanced image processing, such as fast Fourier transform (FFT) analysis, lattice spacing measurement precision reaches ~0.001-0.002 nm. In addition, we measure the sum of the spacing between many lattice fringes and then average it to the adjacent spacing, which will definitely improve the accuracy of the measurement. Such operation makes the detection of 0.003 nm difference meaningful, similar to that reported other where (Nat. Commun.2024,15.1,2159). In the revised manuscript, we re-evaluated the raw data, calculating the value through multiple measurements with a standard deviation of ± 0.001 nm, confirming the 0.003 nm change as statistically detectable.

Reviewer 2: This paper claims that accelerating a catalyst powder in a stream of CO₂/H₂ at a stainless steel target leads to an improvement in the catalytic reactivity by a factor of 3. They claim that selectivity to generate CH₃OH is more than doubled. They claim that this is due to the velocity of the catalyst being 75 m/s impacting the stainless steel. This work is based on a Cu/Al₂O₃ catalyst for the first part.

Comment 2-1: The particles are in a flow of up to 75 m/s and then impact a stainless steel target and they are shaken up every 3 s by tapping. They need to show the apparatus in the paper, not in the SI and provide more visual details about the experiment in the paper on the apparatus. They also need to add a timing diagram so we can see when the flow is present and when it is turned off. Is the flow maintained, or do they shut off the flow after the initial impact? Is this what is shown in figure 2g?

Response: Our description in the previous manuscript may not be clear enough, causing these issues for the reviewers. We have made significant revisions to the manuscript, striving to clarify the apparatus used, experiments, and findings, refer to the comments of reviewers, including:1: Add and Describe Detailed Apparatus Diagram (Figure R3), 2: Clarify Gas Flow and Standard DAR Conditions, 3: Detail Air Hammer Operation, 4: Explain Flow Rate Transition in Figure 3g and Its Purpose.

Figure R3. Schematic show of the reactor used for dynamic activation catalysts for CO₂ hydrogenation.

Comment 2-2: Remove lines 28 to 32 as they are not relevant.

Response: We followed the reviewer's suggestion and removed this sentence in the revision, although we think that this classic quote reflects the essence of our explanation of dynamic activation catalysis. And, we also think the philosophy is the highest level of science.

Comment 2-3: In the center of mass how much is the velocity changed for particles and the gas? With what energy do the catalyst particles hit the surface and how much energy is there to change the properties of the catalyst particles? How much kinetic energy is there on the impact with the stainless steel? Is this enough to change the surface structures of the catalyst?

Response: Only a portion of the kinetic energy is transferred to the catalyst particles and a portion of the kinetic energy of catalyst particles is consumed in collision to make the changes of catalyst

surface structure.

In DAR we described, MD simulations confirm that the reaction gas ($\text{CO}_2/3\text{H}_2$) exits the 0.1 mm nozzle at a line velocity of ~ 452 m/s at flow rate of 360 ml/min, with catalyst particles cyclically colliding with a rigid stainless steel target at ~ 75 m/s (Fig. S1 of manuscript). Based on fluid dynamics simulations (Fluent software), the input power of the DAR by the gas flow is ~ 0.34 W:

The input energy is evaluated using the following equation:

$$P_K (\text{J/s}) = \frac{1}{2}mv^2$$

where " P_K " is the Kinetic power, measured in watts (W, or J/s), " m " is the mass of the Gas in kilograms (kg), and " v " is the Gas velocity, in meters per second (m/s).

Imagine the catalyst particles as copper loaded alumina with a diameter of 20 nanometers and a length of 100 nanometers, with a density of 4 g/cm^3 . In 40wt.% Cu content, a particle contains ~ 13000 Cu atoms. The kinetic energy of one particle in 75 m/s velocity is

$$E_K = \frac{1}{2}\rho\pi r^2Lv^2$$

where E_K is the kinetic energy of a particle, measured in Joule, ρ is the density, r and L is respectively the radius and length of the catalyst nanorod, and v is the particle velocity, in meters per second (m/s). The E is estimated as $\sim 3.5 \times 10^{-16}$ J, i.e., ~ 1840 eV. If the particle is totally stopped at the collision and a half kinetic energy is absorbed by the particle and \sim one over three copper atoms is affected by the collision, the average energy felt by each copper atom is about 0.25 eV. The actual energy should be less than this value. Generally, we concluded that the energy of collision was enough to change the surface structure of copper, but not enough to increase the locale temperature too much, upon the collision.

To further validate this, we employed density functional theory (DFT) to compare the energy difference between the discrete condensed state and the normal Cu structure. The discrete condensed Cu structure was extracted from MD simulation snapshots and fully optimized via DFT (K-POINTS = $5 \times 5 \times 1$, ENCUT = 800 eV). To isolate the energy change of Cu, the support (Al_2O_3) was removed, and the system energy was calculated at higher precisions. The results show that the energy of the discrete condensed Cu is 0.17 eV/atom higher than that of the normal Cu structure, closely matching the estimated input energy (0.25 eV/atom). This indicates that, theoretically, the collision energy in DAR is sufficient to induce a transition of the Cu surface structure to the discrete condensed state.

Additionally, we simulated Cu layers (2–4 atomic layers) supported on Al₂O₃ to calculate the energy required to remove a single Cu atom, using the following formula:

$$E_{\text{vac}} = E_{\text{defect}} + E_{\text{iso}} - E_{\text{perfect}}$$

where E_{vac} is the energy required to completely remove a Cu atom (eV), E_{defect} is the energy after removing a Cu atom from the surface of the complete structure (eV), E_{iso} is the energy of a single Cu atom in a vacuum (eV), and E_{perfect} is the energy of the complete structure (eV).

The calculated vacancy formation energies at the 2nd, 3rd, and 4th Cu layers are 5.27 eV, 5.07 eV, and 5.79 eV, respectively. These energies far exceed the collision energy in DAR, suggesting that mechanical impacts or milling (e.g., in SBMR) are unlikely to completely dislodge Cu atoms from the lattice but are sufficient to elongate Cu-Cu bond lengths or alter the crystalline structure. To explore this, we constructed a model with three Cu layers (60 Cu atoms) on Al₂O₃, generating an amorphous Cu structure via rapid annealing at 300–2500 K, followed by DFT global optimization. Compared to the traditionally supported Cu structure, the amorphous Cu structure exhibits an energy increase of 2.76 eV, which exceeds the input energy of DAR (~0.34 W) but is achievable in the stirred ball mill reactor (SBMR, ~90 W motor power). This indicates that the excessive energy in SBMR disrupts the Cu crystalline structure entirely, whereas the mild collisions in DAR precisely modulate Cu into the discrete condensed state, preserving catalytic activity.

Comment 2-4: I think what they are showing is that the collision with surface changes the morphology. This concept is not new. The way to do it may be new, but the consequences are not new or novel. As this is essentially a mechanical force applied to the surface to distort it, how much force is there is other means of doing this are present. How dependent are the catalyst changes on the flow velocity and hence the impact energy?

Response: Thanks for the comment. In the dynamic activation reaction (DAR), we utilize high-velocity gas flow to carry catalyst particulates to collide cyclically with a rigid target, inducing surface morphology changes that significantly enhance CO₂ hydrogenation to methanol. This reactor does not use any mechanical rotating parts and it is innovative compared with traditional methods like the stirred ball mill reactor (SBMR). To emphasize DAR's uniqueness, we focused on comparisons with traditional thermal catalysis, which may have led to the omission of SBMR experimental results. To address this, we provide supplementary data: SBMR, with an energy input

of ~ 90 W (Power of the motor), caused severe lattice collapse, achieving only $230 \text{ mg}\cdot\text{g}_{\text{cat}}^{-1}\cdot\text{h}^{-1}$ methanol space-time yield, far below DAR's $660 \text{ mg}\cdot\text{g}_{\text{cat}}^{-1}\cdot\text{h}^{-1}$, and exhibited significant deactivation (CO selectivity rising to 75%, see above Figure R1). In contrast, DAR's energy input is milder (a 0.1 mm nozzle jet at an instantaneous velocity of 452 m/s generates ~ 0.34 W of kinetic energy), yet sufficient to induce effective morphological changes. Catalyst transformations strongly depend on flow rate. For example, at $160 \text{ ml}\cdot\text{min}^{-1}$ (Figure S6), no dynamic activation occurs, whereas $360 \text{ ml}\cdot\text{min}^{-1}$ boosts STY nearly threefold, underscoring impact energy as key to performance optimization.

We sincerely appreciate the question and incorporate a discussion on mechanochemistry in line 151 of the revised manuscript to strengthen the theoretical framework of our study.

Comment 2-5: I think that much of what they have done may also be to get more surface area available for the catalyst. How much more surface area is there? Also, as the particles are in the gas phase due to collisions and tapping, how does that affect the surface area?

Response: In the dynamic activation reaction (DAR), the grain size of copper loaded on alumina reduced from 4.04 nm to 3.08 nm and the surface area of catalyst increased from $63.4 \text{ m}^2/\text{g}$ to $69.8 \text{ m}^2/\text{g}$ after reaction. As comparison, it decreased to $56.3 \text{ m}^2/\text{g}$ after traditional reaction.

The main function of tapping the reactor is to use vibration to eliminate the adhesion of catalytic powder particles to the wall. The tapping has no effect on the particle size and surface area of the catalyst.

Comment 2-6: How much of what is happening is just forming more active sites by collision with the surface and any reactions with what is on the steel surface to change particle morphology?

Response: The catalyst composition has little changed EDS and ICP analyses (Figure. S11) detect no Fe, Cr, or Ni from the stainless steel target after the reaction. The collision caused the reduction of Cu grain size and the increase in catalyst surface area. The target's inertness (300°C , 2.0 MPa) and conductivity ($\sim 16 \text{ W}\cdot\text{m}^{-1}\cdot\text{K}^{-1}$) ensure morphological changes arise from mechanical collisions. In the revised manuscript, lines ~ 231 quantify collision contributions and the change of Cu state is the main cause.

Comment 2-7: What is the effective temperature of the system on collision?

Response: This is difficult question to quantitatively answer.

Also refer to the response to Comment 1-1 about hotspot, as listed above, for a 10 nm particle,

the kinetic energy ($\sim 1.32 \times 10^{-17}$ J), if fully converted to heat, could increase the temperature by ~ 27 K. However, the collision duration is extremely brief ($\sim 10^{-10}$ s), and with rapid gas flow (360 ml/min) and the high thermal conductivity of stainless steel (~ 16 W/m·K), heat dissipates quickly, maintaining the system's overall temperature at 300°C.

Characterization results (e.g., XRD, EXAFS, HAADF-STEM) reveal that collisions primarily induce lattice distortion, reduced coordination numbers, and grain size reduction, rather than thermally induced deactivation (e.g., sintering). Mechanochemical literature (Science, 366(6472), 1451-1452.) indicates that localized heating is often negligible in certain mechanically driven systems, consistent with our observations.

Control experiments (Figure R2a, b) in a fixed-bed reactor (FBR) reveal that higher temperatures increase CO₂ conversion but yield predominantly CO ($> \sim 90\%$ selectivity), contrasting with DAR's $\sim 95\%$ methanol selectivity, indicating negligible impact of transient heating on methanol performance. Thus, the effective system temperature during collisions remains 300°C, with performance enhancements driven primarily by structural changes. We will add a discussion of this in line 151 of the revised manuscript to provide a more comprehensive explanation of the dynamic activation mechanism.

Comment 2-8: They need to move Table S2 to the paper.

Response: Thanks for the suggestion. Table S2 has been moved to the manuscript as Table 1 in the revised manuscript.

Comment 2-9: Line 91 'latter' not 'later'.

Response: Thanks, it was fixed in revision.

Comment 2-10: The authors should look at the fast reactor flow work of Lanny Schmidt from the University of Minnesota.

Response: Thanks for the suggestion. Lanny Schmidt's fast reactor flow research at the University of Minnesota is renowned for developing millisecond contact time reactors, which achieve efficient catalytic reactions (e.g., methane partial oxidation to syngas, ethanol to hydrogen) through ultra-short contact times, emphasizing the role of high-velocity gas flow in optimizing selectivity and conversion. Related contents have been added in the revised manuscript and some publication of Lanny Schmidt has been added as references (Science, 1996, 1560-1562.).

Comment 2-11: Error bars are needed for all of Figure 2.

Response: Thanks for the comment and it has been revised accordingly.

Comment 2-12: How reproducible is the work?

Response: The work is highly reproducible.

Comment 2-13: Line 144, remove 'elaborate' as there is nothing special in terms of the characterization. They could change it to extensive.

Response: Thanks for the comment. Revision was made accordingly.

Comment 2-14: Line 193. At least state the type of DFT calculations.

Response: Thanks for the comment. The discrete condensed copper was obtained based on the molecular dynamic (MD) simulations, please see the supporting information. During the MD simulation, one frozen frame was randomly selected and optimized using the Vienna Ab-initio Simulation Package (VASP) software, of which the detailed computational method was shown in the supporting information. The obtained model was used to represent the unsteady state of copper in the dynamic activation reactor (DAR).

Comment 2-15: Masny more details are needed to reproduce the Fluent results. What is given is not enough. There is no information meshing, what parts of Fluent were used, etc.

Response: Thanks for the comment. In the Computational Methods section, we have supplemented detailed simulation settings. The simulation employed double-precision calculations, utilizing hexahedral mesh generated via the ICEM module of Ansys 2022R1, comprising 560,000 cells, with refined mesh density around the target to enhance resolution. The impact process was simulated using the FLUENT module, with catalyst particles modeled via the Discrete Phase Model (DPM). Spatial discretization was performed using the Green-Gauss cell-based method, and flux calculations adopted the distance-based scheme proposed by Rhie-Chow.

Comment 2-16: How was the frequency analysis done for the transition state DFT calculations?

Response: The frequency analysis was conducted over one image generated via the NEB method. According to the frequency of the adsorbates, the transition state was confirmed if only a single imaginary frequency was found, just as presented in Figure R4.

1	f =	85.743313	THz	538.741126	2PiTHz	2860.088985	cm ⁻¹	354.605967	meV
2	f =	49.982811	THz	314.051261	2PiTHz	1667.247048	cm ⁻¹	206.712363	meV
3	f =	44.638538	THz	280.472209	2PiTHz	1488.981323	cm ⁻¹	184.610222	meV
4	f =	36.216296	THz	227.553699	2PiTHz	1208.045566	cm ⁻¹	149.778615	meV
5	f =	21.622653	THz	135.859139	2PiTHz	721.254062	cm ⁻¹	89.424139	meV
6	f =	14.629358	THz	91.918969	2PiTHz	487.982851	cm ⁻¹	60.502184	meV
7	f =	12.335371	THz	77.505421	2PiTHz	411.463669	cm ⁻¹	51.015011	meV
8	f =	8.591053	THz	53.979175	2PiTHz	286.566657	cm ⁻¹	35.529750	meV
9	f =	5.665666	THz	35.598430	2PiTHz	188.986272	cm ⁻¹	23.431320	meV
10	f =	4.778795	THz	30.026053	2PiTHz	159.403430	cm ⁻¹	19.763514	meV
11	f =	1.761771	THz	11.069536	2PiTHz	58.766366	cm ⁻¹	7.286103	meV
12	f/i=	17.162246	THz	107.833572	2PiTHz	572.470891	cm ⁻¹	70.977370	meV

Figure R4. The frequency analysis of H₂CO* over the discrete condensed copper.

Comment 2-17: What type of energy for figure 4? Electronic energy? Free energy?

Response: The type of energy for Figure 4 is the electronic energy, and we normalize the initial state to 0 for a more intuitive comparison

Comment 2-18: How good is the force field used for the MD simulations to get the coordination at Cu correct as well as the bond distances? What does the MD really show? How good are the MD energetics?

Response: To date, no dedicated potential function for Cu supported on Al₂O₃ surfaces has been reported. Consequently, we employed the Modified Embedded Atom Method (MEAM) potential for Cu (e.g., J PHYS CHEM SOLIDS. 2018, 112: 61-72.) and the Embedded Atom Method (EAM) potential for Al₂O₃ (e.g., PRB. 1994, 50(16): 11996.). Due to the weak electrostatic interactions at the Cu/ Al₂O₃ interface, we used the Lennard-Jones (LJ) potential to describe interfacial interactions. Although this approach may introduce minor errors, our focus is on the impact of the discrete condensed state formed on the Cu surface post-collision on catalytic performance, with interfacial Cu-Al₂O₃ interactions having negligible influence on surface Cu structural changes. These errors do not compromise the qualitative analysis of the Cu surface state. The selected Cu MEAM potential has been validated to accurately reproduce physicochemical properties of Cu (e.g., latent heat, melting expansion, liquid structure factor, and solid-liquid interfacial stiffness) in agreement with experimental data. Compared to classical Cu potentials, it provides superior accuracy in calculating elastic constants, heat capacity, and thermal expansion coefficients, ensuring the qualitative reliability of the simulation results.

Molecular dynamics (MD) simulations were conducted to investigate the dynamic state of the Cu surface under reactive gas impact and to elucidate the structure-activity relationship with

catalytic performance. The results reveal that, under high-velocity gas flow, Cu surface atoms colliding with the target exhibit disordered lattice arrangements, with some atoms displaced from equilibrium positions and Cu-Cu bonds elongated but not fully ruptured, forming a discrete condensed state. In the MD simulations, energy minimization was performed with a precision of 1×10^{-8} , and the temperature and pressure control factors for dynamic relaxation were set to 100 and 1000 times the time step, respectively, to ensure reasonable fluctuations in system temperature and pressure. Although MD results exhibit some deviations compared to density functional theory (DFT), MD serves as a valuable tool for deepening mechanistic understanding, offering high accuracy in qualitative analyses, particularly for studying dislocations, grain boundaries, and other structural evolutions in metal particles, and is widely applied in related fields.

Comment 2-19: In the MD simulations, how much higher in energy is the distorted structure than the non-distorted one? How much higher at the DFT level?

Response: As detailed in Comment 2-3, density functional theory (DFT) calculations indicate that the energy of the discrete condensed Cu state is 36.7 meV/atom higher than that of the normal structure. In molecular dynamics (MD) simulations, we compared the energy of the distorted structure at the moment of collision with that of the non-distorted structure relaxed at 300 K. The MD results show that the distorted structure's energy is 52 meV/atom higher than the non-distorted structure. The close agreement between these results suggests that MD simulations provide reliable qualitative insights for comparative analysis.

Comment 2-20: What are two states of the Cu in the DFT simulations? Were the complete DFT structures optimized? If so, why did they not relax to the lowest energy one? How big is the unit cell?

Response: The pristine state copper is constructed using a common supported model, which represent the catalyst in the traditional fixed bed reactor (FBR). As for the discrete condensed copper, it was built from the MD simulation process to describe the unsteady state of copper in the dynamic activation reactor (DAR).

Both structures mentioned in our work were completely optimized. Regarding to the discrete condensed one, as we mentioned in this work, it is a metastable state, so it was selected from one frozen frame during the MD simulation process. Although it was DFT optimized, the energy of

which is not the minimum.

As for the unit cell, the unit cell for model with pristine state is $15 \times 15 \times 38 \text{ \AA}$, while the unit cell for the discrete condensed model is $12 \times 11 \times 40 \text{ \AA}$.

Comment 2-21: The work on the MoS₂ is just tacked on and there are insufficient details to reproduce it or to gain anything from what is there. Remove it and focus on the CuO on Al₂O₃.

Response: Thanks very much for the suggestion. In the revised manuscript, we remove the MoS₂ related results and focus exclusively on the Cu/Al₂O₃ catalyst, providing a more cohesive and detailed exploration of its dynamic activation and catalytic performance in CO₂ hydrogenation.

Comment 2-22: line 246. What is STY?

Response: STY refers to the space-time yield, defined as the mass of methanol produced per unit mass of catalyst per unit time ($\text{mg} \cdot \text{g}_{\text{cat}}^{-1} \cdot \text{h}^{-1}$). In our study, it quantifies the catalytic efficiency of Cu/Al₂O₃ for CO₂ hydrogenation. We include a clear definition of this term in the revised manuscript to enhance readability.

Comment 2-23: Can this process be used for at an industrial level? What is the energy input for the flow and the tapping?

Response: The dynamic activation reactor (DAR) exhibits considerable potential for industrial-scale CO₂ hydrogenation to methanol, utilizing a low power input of $\sim 0.34 \text{ W}$ to drive Cu/Al₂O₃ particles at $\sim 75 \text{ m/s}$ against a rigid target, imparting 0.25 eV per Cu atom to induce a discrete condensed state, significantly enhancing catalytic activity and selectivity. The DAR's mild activation mechanism, coupled with a six-fold increase in methanol space-time yield ($\sim 660 \text{ mg} \cdot \text{g}_{\text{cat}}^{-1} \cdot \text{h}^{-1}$) and $\sim 95\%$ selectivity (Figure. 3), underscores its suitability for continuous processes requiring high-efficiency catalysts.

Industrialization necessitates optimization of reactor design, catalyst recycling, and gas flow stability to ensure consistent particle-target collisions and mitigate long-term catalyst wear. On large industrial apparatus, the tapping may not be necessary, for it serves solely to prevent particle adhesion in lab apparatus, without direct involvement in catalysis. Flow energy input essentially comes from gas compressors.

Response to the reviewers' comments

We really thank the reviewers for their instructive comments. We have made revision of the manuscript, considering these comments and suggestions. All changes have been highlighted in yellow in the revised manuscript and supporting information. The point-to-point responses to the reviewers' questions are attached bellow.

Reviewer 1: The manuscript and SI are substantially improved. I am not sure that I agree with all of the authors' conclusions, but I think that publication is warranted so the research community can see this work and comment on it or critique it.

Response: We sincerely thank the reviewer for the positive evaluation and recommendation for publication, as well as for acknowledging the substantial improvements in the manuscript and Supplementary Information (SI). We have incorporated into the revisions the issues previously raised by the reviewer concerning velocity changes, the impact of energy on catalyst surface structure alterations, local effective temperatures, and molecular dynamics simulation energy types and validation, to further strengthen the research outcomes. (cf. Revised manuscript: lines 49-58; 155-165; 239-247; 285-292; Figs. 1-3. SM: lines 113-158; Figs. S4, S9, Table S3. etc)

Comment 1-1: I have one additional request. The authors provide excellent information in the response to the review. Please make sure that all of the responses are at least in the SI. include responses to comment 2-3, comment 2-7, and comment 2-18.

Response: Thanks for the specific guidance. We have added all reviewers' comments and our corresponding responses into the Supplementary Information (SI) for transparency and completeness, including the detailed responses to Comment 2-3 (on velocity changes, impact energy sufficiency, and surface structure alterations. Lines 239-247, Page 11), Comment 2-7 (on effective collision temperatures. Lines 155-165, Page 8), and Comment 2-18 (on MD simulation energy types and validation. SI-Lines 113-158, Page 5).

Comment 1-2: They should also provide some comment on the potential for practical use of this approach at industrial scale. See comment 2-23.

Response: Thanks for the significant issues about the future of the dynamic activation reactor

(DAR). Based on present results, 0.66 g/g_{cat}/h, in concepts, for production of a 10,000-ton/year methanol, only ~1.8 tons of catalyst would suffice. In nature, the DAR is also a fluidized-bed reactor with a special collision and reaction zone. Structurally, it is a combination of a gas flow mill and a cyclone separator. Through structural optimization design, its volume can definitely be designed very small. Of course, if we really go to do such an industrial process, we will definitely use better catalysts and higher value reactions. For example, for the reaction of CO₂ hydrogenation, the DAR loaded with MoS₂ catalytic material shows much better catalytic performance and long-term stability. Some sentences are added to the revised manuscript (Lines 285-292). Thanks very much again.

Reviewer 3: Ding et al. report on a highly unusual reactor set-up, which uses a rapid stream of CO₂/H₂ to mechanically mix Cu/Al₂O₃ catalyst particles, resulting in greatly increased activity and selectivity for methanol, compared to conventional thermocatalysis. They provide an explanation in the form of mechanochemical activation of the Cu particles, caused by impacting on the reactor backstop at high velocity, resulting in altered changed Cu nanoparticle structure. However, they are unable to convincingly demonstrate this changed structure, relying on statistically insignificant differences in XRD and TEM data. Computational explorations are provided, however these simulations amplify the alleged differences between the activated and unactivated catalyst structures, reducing the reliability of these results. Previous criticisms of insufficient detail to reproduce results have been adequately dealt with (to the best of my knowledge). The authors have certainly discovered an interesting phenomenon, but have not been able to convincingly explain it.

Response: We thank the reviewer for recognizing such an intriguing phenomenon in our DAR system. Regarding the computational concerns, our simulations employ representative structures post-collision, as we posit that reactions predominantly occur on the high-activity states generated during impacts. However, not all catalyst particles are simultaneously in this activated state; many reside in non-colliding or descending phases within the reactor, thereby diluting the overall activity expression. The selected local collision structures enable qualitative exploration of the activity enhancement mechanism, deepening our understanding of processes within the impact reactor. Furthermore, molecular dynamics simulations yielded a metastable structure, upon which DFT calculations were performed to compare the relative barriers of the RWGS and methanol synthesis

pathways, aligning well with experimental observations. We thank the reviewer supports our research methodology.

Comment 3-1: It seems unlikely to me that the velocity numbers added to the abstract would be well understood by readers in the field of heterogeneous catalysis.

Response: Thanks for the suggestion. We have already revised the abstract section and removed the relevant description (Lines 11-15, Page 1).

Comment 3-2: Line 36: “to precisely activate the catalyst surface” – I don’ t believe there is anything chemically “precise” about this reactor (or mechanism of activation), and would suggest an alternative wording.

Response: Thanks for your suggestion, and we have replaced " precisely " with "effectively".

Comment 3-3: Line 43: Djéga-Mariadassou and Boudart, 2003 (Reference 7) state: “The QSSA theory is based on three fundamental rules: (i) concentrations of intermediates are very low; (ii) the variation of the concentration of one intermediate is stationary, i.e., independent of time ($d[\text{intermediate}]/dt=0$); (iii) consequently, rates of all steps in a sequence have the same value, once divided by the stoichiometric number σ_i , the number of times we need an i-elementary step in the sequence to obtain the overall chemical equation.” With respect to the line in this work: “challenging the traditional notion about the relationship of reaction rate with space velocity”, I would content that when the assumptions of Djéga-Mariadassou and Boudart are sufficiently perturbed, new behaviour would not be surprising, but expected. Could the authors comment further on the differences between their system and the ‘traditional’ kinetic analysis, and specifically which assumptions need to be reevaluated?

Response: We thank the reviewer for highlighting the seminal framework of Djéga-Mariadassou and Boudart (2003 Reference 7), which delineates the quasi-steady-state approximation (QSSA) through its core tenets. We concur that substantial disruptions to these postulates may engender foreseeable departures from classical kinetics, a perspective that resonates with the anomalous space-velocity dependence observed in our dynamic activation reactor (DAR).

Distinct from conventional analyses predicated on static catalyst surfaces (e.g., fixed-bed

reactors), the DAR introduces mechanochemical perturbations via high-velocity particle collisions, engendering a perpetually evolving interface with transient active sites (e.g., Discrete condensed state gifted distorted and elongated lattice). This paradigm principally interrogates:

Assumption (ii): The presumption of stationarity is invalidated by fleeting structural metamorphoses, corroborated by the ~ 2 h relaxation timescale in gas-hourly-space-velocity switching assays (Fig. 3g), thereby engendering temporally heterogeneous intermediate kinetics.

Assumption (iii): Alterations in the reaction pathway—exemplified by diminished activation energies for CO hydrogenation in density functional theory computations (Fig. 5)—disrupt rate parity, preferentially channeling flux toward methanol synthesis over the reverse water-gas shift.

In revisions, we further elaborate on this contrast in lines 49-58 of page 3. Thanks again.

Comment 3-4: Fig. 2 is too complex to readily absorb the information on the reactor design. Fig. 2 should be reproduced in clear, schematic form, with the reactor components properly sized and labelled. For example, there is no need to represent a GC in this figure.

Response: Thanks for the feedback. In revisions, we simplify the Fig. 2 into a concise schematic with a more straightforward presentation, accurately labeling the positions and names of reactor components, and removing extraneous elements.

Comment 3-5: What is the catalyst particle size? How does this change after the reaction? How are the catalyst particles prevented from being blown through the reactor?

Response: We conducted particle size analysis on the catalyst before and after the reaction, using laser scattering particle size analyzer (Mastersizer 2000). It has been added in revised Supplementary Materials as Fig. S4. Before reaction, the catalyst particle size exhibits a bimodal distribution with one less than 100 nm and the other at ~ 100 μ m. Considering the catalyst preparation process, the catalyst should be viewed as micron particle composed of nanoparticles. After reaction, the catalyst particle size exhibits a unimodal distribution centered at ~ 45 μ m, revealing that collisions reshape the catalyst and the catalyst particulates are constantly reorganizing and changing. By images of TEM observations, however, the nanoparticles or the nano domains of the catalyst particles not show significant changes. But the Cu nanoparticles loaded on alumina support changed from the initial average diameter of 4.04 nm to 3.08 nm after the dynamic activation

reaction (Fig. 4, Fig. S11). In contrast, after the traditional fixed-bed reaction (FBR), the size of Cu increases to 6.81 nm due to agglomeration (Fig. S12).

Fig. S4. Catalyst particle size distribution measured using laser scattering particle size analyzer before and after DAR reaction.

In Fig. S2, we illustrate the flow path of the reaction gas. In the expansion section of reactor up part, the velocity of reaction gas is only ~ 2 cm/s and the secondary particles of catalyst in fact move to the inner wall of the reactor, further loss of speed, and slide down to the high-speed flow area for the next collision-reaction cycle. In addition, the strainer (quartz wool) at the outlet of the reactor further prevents the particles from escaping.

Comment 3-6: The STYs in Figs. 3b, 3c and 3d should be in $\text{mmol g}_{\text{cat}}^{-1} \text{h}^{-1}$, so that the species can be compared directly.

Response: Revision has been made accordingly, as shown in Figure 3 of revised manuscript. Thanks.

Comment 3-7: Line 186: “the coordination number of Cu-Cu for the sample under dynamic activation is clearly reduced ($9.4 \pm 0.9 \rightarrow 8.4 \pm 0.8$)” – these numbers are the same, within error. The supposed Cu-Cu bond elongation ($2.535 \text{ \AA} \rightarrow 2.539 \text{ \AA}$) is also remarkably small – these numbers require an associated error (errors are given in Table R1 which suggest that the distances are identical). The difference appears to be far below the resolution limit of XRD (which is closer to 1 \AA than 0.01 \AA , depending on the X-ray frequency).

Response: Thanks for the thoughtful question. It should be noted that due to the characteristics of the reaction process, operando characterization is difficult to carry out and these characterizations

were performed on the samples after the reaction, and the structural changes of them have already undergone relaxation. To ensure EXAFS reliability, we reanalyzed the data, as detailed in our prior response. Initially, the amplitude reduction factor (S_0^2) was referenced from Cu foil and fitted independently per sample. In the revised fitting, we fixed Cu foil CN at 12 to determine S_0^2 , holding it constant across dynamic, static, and fresh samples for precise CN quantification. Final fits satisfy constraints: $\Delta E_0 < 10$ eV, first-shell $\sigma^2 < 0.01$ Å², and $\Delta R < 0.1$ Å, yielding refined values: CN = 9.7 ± 0.7 to 7.8 ± 0.7 , bond lengths = 2.535 to 2.538 Å. We acknowledge the reported CN reduction lies within initial error margins, and the Cu-Cu elongation (2.535 to 2.539 Å) is subtle, potentially below perceived resolution thresholds. These metrics indeed require stringent validation, as direct operando measurements under dynamic conditions remain challenging due to the metastable activated state, which largely relaxes post-reaction (~2 h, revised Fig. 3g), leaving only residual traces.

As to XRD refinement, instrument resolution indeed exceeds refinement errors, but Rietveld refinement is not a direct bond measurement. Instead, it amplifies subtle signals influencing peak intensity to reflect minute internal crystal parameters, with bond length uncertainty in powder data ~0.002 Å (*Powder. Diff.*, 2007, 74-82). A 0.008 Å change surpasses this error; moreover, our samples—relaxed for more than 2 h post-reaction—further narrow the gap between post-reaction and fresh catalysts, yet still facilitate analysis of impact-induced Cu-Cu structural shifts. Referenced to Table R1 listed below, hope the reviewer supports our viewpoints. Revised related sentences have been marked in lines 192-197 of page 10. Tables S3 listed Cu K-edge EXAFS data of the catalysts was also revised accordingly. Thanks again.

Table S3. Cu K-edge EXAFS data for the catalysts ^a.

Samples	Shell	R (Å) ^b	CN ^c	σ^2 (Å ²) ^d	ΔE_0 (eV)	r-factor
40Cu-fresh	Cu-Cu	2.535±0.005	9.7±0.7	0.0075	2.27	0.009
40Cu (FBR)-used	Cu-Cu	2.534±0.006	9.8±0.8	0.0079	2.05	0.011
40Cu (DAR)-used	Cu-Cu	2.538±0.005	7.8±0.7	0.0085	2.41	0.009
Cu foil	Cu-Cu	2.537±0.007	12	0.009	5.18	0.012

a: The S_0^2 value are 0.869 according to the experimental EXAFS fit of Cu foil by fixing CN as the known crystallographic value for Cu K-edge EXAFS spectra fitting, respectively; *b*: bond length; *c*: coordination number; *d*: Debye-Waller factor.

Comment 3-8: Similarly, the lattice fringe distance has a quoted error of only 0.001 nm (i.e. 0.01 Å), which is unbelievably small

Response: Thanks for noting the quoted error of ± 0.001 nm (± 0.01 Å) in lattice fringe distances (e.g., Cu(111) from 0.208 to 0.211 nm via HAADF-STEM). While seemingly minute, this precision is attainable with aberration-corrected STEM, where sub-Ångström resolution (~ 0.05 nm or better) and fast Fourier transform (FFT) analysis of multiple fringes enable ~ 0.001 nm accuracy, as validated in nanomaterial studies (e.g., *Nat. Commun.* 2020, 11, 6095). In revisions, we have reassessed the raw data, deriving a standard deviation of ± 0.001 nm from repeated measurements, confirming the 0.003 nm change is statistically significant (Lines 200-206, Page 10). Thanks again.

Comment 3-9: Of concern is the discrepancy between the Cu-Cu distances and the CNs in the text vs in Table R1 (particularly the much lower CN in the table than in the text). Which of these is the correct value?

Response: We thank the reviewer for highlighting this discrepancy in the Cu-Cu distances and coordination numbers (CNs) between the main text and Table R1. The values in Table R1 represent the correct, refined results from our reanalysis, employing a standardized amplitude reduction factor ($S_0^2 = 0.87$) fixed across samples for enhanced precision and consistency. The text values were from an earlier fitting iteration and have not yet been synchronized. We update the manuscript text accordingly in revisions to reflect these accurate figures, ensuring alignment and statistical

robustness. Thanks very much for the careful reading.

Comment 3-10: Previous Reviewer's Comment 2-4 is not sufficiently engaged with – I agree with the previous reviewer that “How dependent are the catalyst changes on the flow velocity and hence the impact energy?” is a key question that remains unanswered.

Response: We thank the reviewer for reiterating this critical query from previous Comment 2-4, emphasizing the dependence of catalyst changes on flow velocity and impact energy—a pivotal aspect for elucidating the DAR mechanism. We concur and apologize for insufficient prior engagement; our initial focus on contrasting DAR with thermal catalysis inadvertently underemphasized this dependency, while highlighting innovations like the absence of mechanical parts.

To clarify, DAR leverages high-velocity gas flow to propel catalyst particulates into cyclic collisions with a rigid target, inducing morphological alterations (e.g., reduced Cu particle size from 4.04 to 3.08 nm, inhibiting sintering) that amplify CO₂ hydrogenation performance. This is distinct from traditional mechanochemical approaches like stirred ball milling (SBMR, ~90 W input), which cause severe lattice collapse, yielding inferior methanol space-time yield (230 mg·g_{cat}⁻¹·h⁻¹ vs. DAR's 660 mg·g_{cat}⁻¹·h⁻¹) and rapid deactivation (CO selectivity increases to ~75%, cf. Fig. S9 of revised Supplementary Materials). DAR's milder energy (~0.34 W from a 0.1 mm nozzle jet at 452 m/s) enables non-destructive activation.

Catalyst transformations exhibit strong flow-rate dependence: at 160 ml·min⁻¹ (Fig. S7), impact energy remains sub-threshold (below ~0.25 eV/Cu atom) and yields no activation with FBR-like performance; at 360 ml·min⁻¹, energy reaches the threshold sufficient for structural shifts, boosting yield nearly threefold. This underscores a critical energy threshold (~0.25 eV/Cu atom, per DFT), tied to Cu lattice dynamics, for optimizing activation without degradation. The relevant content has been incorporated into the revised manuscript (Lines 239-247, Page 12).

Comment 3-11: Previous Reviewer's Comment 2-12 has not been adequately answered – how have the error bars in the new Fig. 3 been calculated?

Response: Thanks for the careful reading. The error bars in the revised Fig. 3 represent the standard deviation calculated from three independent experiments under identical conditions. Each replicate

used fresh catalyst batches, and it has been detailed in the revision (Lines 174-175, Page 10).